# Pipecolisporin, a Novel Cyclic Peptide with Antimalarial and Antitrypanosome Activities from a Wheat Endophytic *Nigrospora oryzae*

**DOI:** 10.3390/ph14030268

**Published:** 2021-03-16

**Authors:** Ignacio Fernández-Pastor, Victor González-Menéndez, Frederick Annang, Clara Toro, Thomas A. Mackenzie, Cristina Bosch-Navarrete, Olga Genilloud, Fernando Reyes

**Affiliations:** 1Fundación MEDINA, Centro de Excelencia en Investigación de Medicamentos Innovadores de Andalucía, Parque Tecnológico de Ciencias de la Salud, Avda. del Conocimiento 34, 18016 Granada, Spain; ignacio.fernandez@medinaandalucia.es (I.F.-P.); victor.gonzalez@medinaandalucia.es (V.G.-M.); freddie.annang@medinaandalucia.es (F.A.); clara.toro@medinaandalucia.es (C.T.); thomas.mackenzie@medinaandalucia.es (T.A.M.); olga.genilloud@medinaandalucia.es (O.G.); 2Instituto de Parasitología y Biomedicina “López-Neyra”, Consejo Superior de Investigaciones Científicas (CSIC) Avda. del Conocimiento 17, Armilla, 18016 Granada, Spain; cristinabosch@ipb.csic.es

**Keywords:** pipecolisporin, *Nigrospora oryzae*, fungal endophyte, structural elucidation, Marfey’s analysis, antiparasitic activity

## Abstract

A novel cyclic antimalarial and antitrypanosome hexapeptide, pipecolisporin (**1**), was isolated from cultures of *Nigrospora oryzae* CF-298113, a fungal endophyte isolated from roots of *Triticum* sp. collected in a traditional agricultural land of Montefrío, Granada, Spain. The structure of this compound, including its absolute configuration, was elucidated by HRMS, 1-D and 2-D NMR spectroscopy, and Marfey’s analysis. This metabolite displayed interesting activity against *Plasmodium falciparum* and *Trypanosoma cruzi*, with IC_50_ values in the micromolar range, and no significant cytotoxicity against the human cancer cell lines A549, A2058, MCF7, MIA PaCa-2, and HepG2.

## 1. Introduction

Mutualism between plants and fungal endophytes is almost universal; indeed, no study has yet reported the existence of any plant species without endophytes [1,2]. Endophytes can be a double-edged sword, as they usually act to protect the host against other phytopathogens, but they may act as opportunistic pathogens during plant senescence or under stress conditions [3,4,5]. Although an enormous diversity of endophyte metabolites and their biological activities have so far been reported, only about 10% of the approximately one million known terrestrial endophytes have been investigated [6]. 

According to the WHO report in 2020, vector-borne diseases account for more than 17% of all infectious diseases, causing more than 700,000 deaths annually. Protozoan parasites, such as *Plasmodium falciparum* or *Trypanosoma cruzi*, are the cause of malaria and Chagas disease, respectively, and it is estimated that approximately a billion people around the world suffer from these diseases every year [7].

Although historically plants have been the main source of antiplasmodial natural products [8], many antimicrobial secondary metabolites from fungal endophytes have also been described as potential antiprotozoal agents. These include, among others, altenusin [9], xylariaquinone A [10], codinaeopsin [11], fusaripeptide A [12], strasseriolides A–D [13], leucinostatins [14], griseofulvin [15], or trichodermin [16]. Other interesting biological properties exhibited by endophytic metabolites include antifungal, anti-inflammatory, antiproliferative, or anticancer activities [17].

The increasing resistance of parasites to conventional drugs makes the research of new and more specific bioactive molecules mandatory, and the discoveries of venturamide A [18], cyclosporin A derivatives [19], and kakeromamide B [20] are examples of this. Cyclic peptides possess several favorable pharmacological features as antiparasitic natural products [21] and their solid-phase synthesis offers a unique advantage for further optimization of the initial hits guided by medicinal chemistry [22].

As a result of the LC/MS survey of new fungal metabolites in extracts from cultures of crop endophytes, we identified an intense peak with the not previously reported molecular formula C_37_H_53_N_7_O_6_ in a culture in solid rice-based medium (BRFT) of the strain CF-298113 of *Nigrospora oryzae*, isolated from roots of *Triticum* sp. collected in a traditional cropland in Spain. The fungus *N. oryzae* is a common endophyte in plants and algae, and only a few natural products have been previously reported from this species. The most remarkable ones are the antibacterial nigrosporins A and B [23], quercetin monoglycosides [5], and the cyclohexadepsipeptides oryzamides A–E [24]. The novelty of the new molecule identified in our *N. oryzae* strain together with the scarce number of compounds isolated from this fungal species prompted us to investigate its chemical structure and biological properties.

## 2. Results and Discussion

### 2.1. Isolation and Structural Elucidation of Pipecolisporin

Fractionation of the methyl ethyl ketone (MEK) extract of a scaled-up culture of *N. oryzae* CF-298113 in BRFT medium (Appendix A), using reversed-phase C18 medium pressure chromatography followed by semipreparative reversed-phase HPLC on a phenyl column, resulted in the isolation of pipecolisporin (**1**) (Figure 1).

Compound **1** was obtained as a white amorphous solid. A molecular formula C_37_H_53_N_7_O_6_ was deduced from its (+)-ESI-TOF analysis ([M + H]^+^ 692.4152, Δ +3.2 ppm) (Appendix A). NMR spectroscopy was extensively applied to determine the structure of the peptide. The ^1^H spectrum of **1** (Appendix A) exhibited characteristic signals for multiple exchangeable protons (δ_H_ 7.0–11.0 ppm), six signals of α-hydrogens of amino acids (δ_H_ 3.8–4.6 ppm), and different alkyl (δ_H_ 0.6-3.8 ppm) or aromatic (δ_H_ 6.9–7.6 ppm) side chain hydrogens. Analysis of COSY and TOCSY spectra (Appendix A, respectively) allowed the identification of spin systems consistent with the presence of the following proteinogenic amino acids: Trp (×1), Ile (×1), Leu (×1), and Pro (×1). It also accounted for the presence of one β-Alanine residue (β-Ala) and one pipecolic acid residue (Pipe). The cyclic nature of the peptide and the connectivity between amino acids was confirmed by the key NOESY and HMBC correlations shown in Figure 2, additionally supported by HRMS/MS fragmentation experiments (Figure 3).

Marfey’s analysis allowed the determination of the absolute configuration of all the amino acid residues as L. For this aim, the retention time of L and D standards of all the amino acids present in the peptide derivatized with L-FDVA, namely Pro, Ile, Leu, Trp, and Pip, were compared with the retention time of a hydrolyzed and derivatized aliquot of **1** (Appendix A). Additionally, due to the coelution of L-Ile and L-*allo*-Ile in the analytical LC/MS conditions employed, the presence of L-Ile in the structure was confirmed by comparison of the HSQC NMR spectra of a hydrolysate of **1** with those of L-Ile and L-*allo*-Ile standards (Appendix A), a strategy recently employed in the determination of the absolute configuration of Ile residues in cacaoidin [25]. 

### 2.2. Biological Activity

The isolated compound was tested against several tropical parasites and exhibited activity in the micromolar range against *P. falciparum* 3D7 and *T. cruzi* Tulahuen C4 parasites. The activity versus the *T. cruzi* Tulahuen C4 parasites was the most remarkable, with a measured IC_50_ of 8.46 μM, comparable to that of the standard drug benznidazole, currently used in the treatment of Chagas disease (IC_50_ in the same assay of 2.21 μM) (Appendix A). The activity measured against *P. falciparum* was also in the micromolar range, with an IC_50_ of 3.21 μM (Appendix A). When tested against the human cancer cell lines A549 (lung carcinoma), A2058 (metastatic melanoma), MCF7 (breast adenocarcinoma), MIA PaCa-2 (pancreatic carcinoma), and HepG2 (hepatocyte carcinoma) by means of a cell viability MTT assay, the compound was found to not be cytotoxic at the highest concentration tested of 50 μM (Appendix A). Additionally, the compound was tested against a panel of microbial human pathogens, including Gram positive (methicillin-resistant *Straphylococcus aureus*, MRSA), Gram negative (*Acinetobacter baumannii*, *Eschericia coli*, and *Pseudomonas aeruginosa*), yeast (*Candida albicans*), and fungi (*Aspergillus fumigatus*), and proved to not be active against any of them at a concentration of 128 μg/mL. 

## 3. Materials and Methods

### 3.1. General Experimental Procedures

Solvents employed were all HPLC grade. Optical rotations were measured on a Jasco P-2000 polarimeter (JASCO Corp., Tokyo, Japan). IR spectra were recorded with a JASCO FT/IR-4100 spectrometer (JASCO Corp.) equipped with a PIKE MIRacle^TM^ single reflection ATR accessory. LC-UV-LRMS analyses were performed on an Agilent 1260 Infinity II (Agilent Technologies, Santa Clara, CA, USA) single quadrupole LC-MS system. HRESIMS and MS/MS spectra were acquired using a Bruker maXis QTOF mass spectrometer (Bruker Daltonik GmbH, Bremen, Germany) coupled to an Agilent 1200 Rapid Resolution HPLC. The mass spectrometer was operated in positive and negative mode. Medium pressure liquid chromatography (MPLC) was performed on a CombiFlash Teledyne ISCO Rf400x apparatus (Teledyne ISCO, Lincoln, NE, USA) with a preloaded C18 column (Phenomenex, 50 µm). Preparative or semi-preparative HPLC purifications were performed on a Gilson GX-281 322H2 HPLC (Gilson Technologies, Middleton, WI, USA) using reversed-phase semi-preparative (XBridge, Phenyl, 10 × 150 mm, 5 um) column. 1-D and 2-D NMR spectra were recorded at 297 K on a Bruker Avance III spectrometer (500 and 125 MHz for ^1^H and ^13^C, respectively) equipped with a 1.7-mm TCI MicroCryoProbe^TM^ (Bruker Biospin, Fällanden, Switzerland). ^1^H and ^13^C chemical shifts were reported in ppm using the signals of the residual solvents as internal reference (δH 2.50 and *δ*_C_ 39.52 ppm for DMSO-*d_6_*; δ_H_ 4.75 ppm for D_2_O). 

### 3.2. Microbial Isolation and Identification

The producer strain CF-298113 was isolated from surface disinfected root pieces of *Triticum* sp. collected in a traditional agricultural land in Montefrío (Granada, Spain) following a previously described standard indirect technique [26]. The axenic strain was preserved as frozen suspensions of septate mycelium and conidia in 10% glycerol at −80 °C. This strain is currently maintained in the Fungal Culture Collection of Fundación MEDINA (http://www.medinadiscovery.com). DNA extraction, PCR amplification, and DNA sequencing were performed as previously described [26]. Sequences of the complete ITS1-5.8S-ITS2 and initial 28S region or independent ITS and partial 28S rDNA were compared with those deposited at GenBank or the NITE Biological Resource Center (http://www.nbrc.nite.go.jp/) by using the BLAST application [27,28]. Database matching with the ITS rDNA sequence (www.fungalbardcoding.org) yielded a complete sequence similarity (100%) to the strain of *Nigrospora oryzae* ATCC 12,772 GenBank Accession No. KU933443), thus indicating that strain CF-298113 was genetically similar to *N. oryzae*, and conspecific. High similar scores to other authentic fungal strains of this species, e.g., *N. oryzae* CBS 113884 [29] (GenBank Accession No. DQ219433, 100% sequence similarity), or *N. oryzae* CBS 231.32 (GenBank Accession No MH855300, 99% sequence similarity) indicated that CF-298113 can be classified as *Nigrospora oryzae* (Berk. & Broome).

### 3.3. Fungal Solid-State Fermentation (SSF)

Strain CF-298113 was fermented by inoculating 10 mycelial agar plugs into SMYA medium (Bacto neopeptone 10 g; maltose 40 g; yeast extract 10 g; agar 3 g; H_2_O 1 L) in two flasks (50 mL of medium in a 250-mL Erlenmeyer). The flasks were incubated on a rotary shaker at 220 rpm at 22 °C with 70% relative humidity. After growing the seed stage for 7 days, aliquots of 4 mL were used to inoculate 20 × 500-mL Erlenmeyer flasks containing the production solid rice-based medium (BRFT). The BRFT medium contained 20 g of brown rice and 40 mL of a solution of yeast extract 1 g/L, sodium tartrate 0.5 g/L and KH_2_PO_4_ 0.5 g/L per flask [30]. The 20 flasks seeded were incubated under static conditions at 22 °C and 70% relative humidity for 21 days. The resulting 20 solid flask fermentaions were extracted by adding MEK (20 × 100 mL) and 20 mL of HPLC quality water to each flask. Then, the flasks were shaken in a Kühner at 220 rpm for 1 h. After that, the mixtures were centrifuged (10 min, 9000 rpm) and filtered under vacuum. All solutions were pooled into a sepation funnel and the organic extract was rotary evaporated until dry (40 g). 

### 3.4. Isolation and Identification of Pipecolisporin

The solid residue obtained was mixed with reversed phase C-18 silica gel in a 1:2 ratio and loaded onto a C-18 column (200 × 35 mm) that was eluted with a stepped H_2_O-CH_3_CN gradient (18 mL/min; 0–100% CH_3_CN in 55 min; UV detection at 210 nm and 280 nm). Those fractions containing **1**, as confirmed by LC/MS analysis, were re-purified by semi-preparative RP-HPLC (X-Bridge, Phenyl, 10 × 150 mm, 5 um) applying isocratic elution of H_2_O-CH_3_CN as mobile phase (3.6 mL/min; 32% CH_3_CN for 40 min; UV detection at 210 nm). Fractions containing the peak eluting at 18–20 min were pooled, the organic solvent was evaporated under a N_2_ stream, and the resulting aqueous solution was freeze-dried. Pipecolisporin (**1**) was thus obtained as an amorphous white powder (10 mg). 

*Pipecolisporin* (**1**): white amorphous solid; [α]_D_ -33.0 (c 0.013, MeOH); UV (DAD) 210, 290 nm; IR (ATR) ν_max_ 3344, 2936, 2871, 1644, 1539, 1443, 1263, 1024 and 743 cm^−1^, ^1^H and ^13^C NMR data see Table 1, (+)-ESI-TOF *m/z* 692.4152 [M + H]^+^ (calcd for 692.4130), 709.4417 [M + NH_4_]^+^ (calcd for 709.4396) and 1400.8444 [2M + NH_4_]^+^ (calcd for 1400.8453).

#### Marfey’s Analysis of Compound **1**

A sample (150 µg) of compound **1** was dissolved in 0.3 mL of 6N HCl containing 5% of thioglycolic acid and heated at 110 °C for 16 h. The resulting mixture was dried overnight under a N_2_ stream. The solid residue was resuspended in 50 µL of a 1 M NaHCO_3_ solution that turned violet. Once the solution stopped fizzing, 150 µL of L-FDVA (Marfey’s reagent, *N*-(2,4-dinitro-5-fluorophenyl)-L-valinamide) was added and the solution turned yellow. The reaction mixture was heated at 40 °C for 40 min till the solution turned red. After that time, the reactions were quenched by dropwise addition of 1N HCl up to the mixture turned yellow again. For the HPLC analysis, 10 µL of the derivatives solution were added to 40 µL of acetonitrile and analyzed by LC/MS on an Agilent 1260 Infinity II single quadrupole LC/MS instrument. Separations were carried out on a Waters X-Bridge C18 column (4.6 × 150 mm, 5 um) maintained at 40 °C. A mixture of two solvents, A (10% acetonitrile, 90% water) and B (90% acetonitrile, 10% water), both containing 1.3 mM trifluoroacetic acid and 1.3 mM ammonium formate, was used as the mobile phase under a linear gradient elution mode (25−65% B in 28 min, 65–100% B in 0.1 min, then isocratic 100% B for 4 min) at a flow rate of 1.0 mL/min. The amino acids standards were derivatizated and analyzed following the same methodology described for compound **1.** HPLC traces of these analyses are shown in Appendix A. Retention times (min) for the observed peaks in the HPLC trace of the L-FDVA-derivatized amino acids standards were as follows: L-Pro: 7.12, β-Ala: 8.02, D-Pro: 9.15, D-Pipe: 11.9, L-Trp: 12.10, L-Ile – L-Leu: 12.10–12.29, L-Pipe: 13.22, D-Trp: 15.2, D-Ile: 18.5, D-Leu: 18.9. Retention times (min) for the observed peaks in the HPLC trace of the L-FDVA-derivatized hydrolysis product of compound **1** were as follows: L-Pro: 6.98, β-Ala: 8.02, L-Trp: 12.10, L-Ile – L-Leu: 12.10–12.29, and L-Pipe: 13.22.

### 3.5. Biological Assays

#### 3.5.1. *Plasmodium Falciparum* 3D7 Lactase Dehydrogenase in Vitro Assay

The IC_50_ of pipecolisporin (**1**) was determined in the *P. falciparum* 3D7 lactate dehydrogenase whole parasite assay as previously described [31]. The isolated compound was tested in duplicate on two different occasions to confirm the IC_50_. 

#### 3.5.2. Transgenic *Trypanosoma cruzi* β-D-galactosidase in vitro Assay

The IC_50_ of **1** was determined in the transgenic *T. cruzi* β-d-galactosidase assay as previously described [32]. The isolated compound was tested in duplicate on two different occasions to confirm the IC_50._


#### 3.5.3. MTT/Cell Viability Assay 

The cell viability of five different human cancer cell lines was studied based on the MTT (3-(4,5-dimethylthiazol-2-yl)-2,5-diphenyltetrazolium bromide) assay [33]: A549 (lung carcinoma), A2058 (metastatic melanoma), MCF7 (breast adenocarcinoma), MIA PaCa-2 (pancreatic carcinoma), and HepG2 (hepatocyte carcinoma). IC_50_ of **1** was determined as previously described [34]. The isolated compound was tested as a 12-point dose–response curve (½ serial dilutions) starting at a concentration of 50 µM in triplicate.

#### 3.5.4. Antimicrobial Assays 

Antimicrobial activity against a panel of human pathogens including MRSA, *A. baumannii*, *E. coli*, *P.aeruginosa, C. albicans*, and *A. fumigatus* was performed as previously reported [35]. The isolated compound was tested as a 10-point dose–response curve (½ serial dilutions) starting at a concentration of 128 μg/mL in triplicate.

## 4. Conclusions

Pipecolisporin (**1**), a new pipecolic acid containing hexapeptide, was isolated from cultures of the endophytic fungus *N. orizae*. The compound displayed remarkable antiparasitic activity against *P. falciparum* and *T. cruzi,* with an IC_50_ value comparable to that of benznidazole, currently used in the treatment of Chagas disease, and no toxicity against a panel of five human carcinoma cell lines. Therefore, the newly isolated peptide can be proposed as a viable starting point for further investigation in vivo towards its potential application in anti-Chagas chemotherapy. 

## Figures and Tables

**Figure 1 pharmaceuticals-14-00268-f001:**
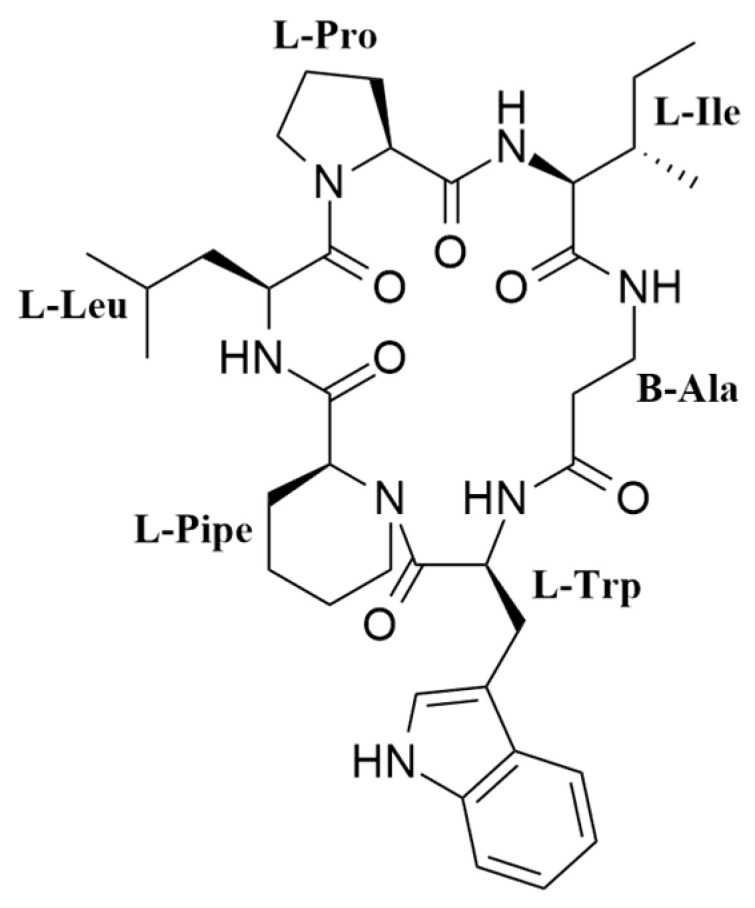
Structure of pipecolisporin (**1**).

**Figure 2 pharmaceuticals-14-00268-f002:**
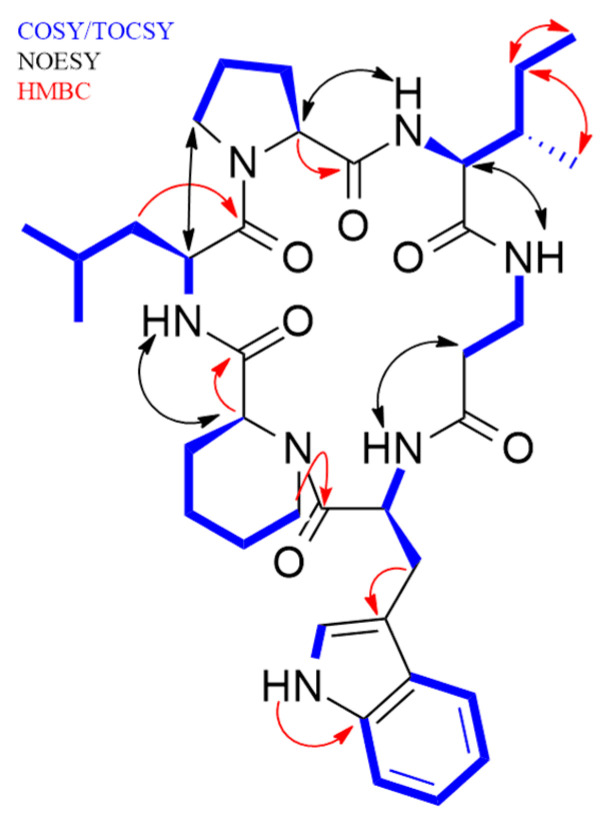
Key ^1^H-^1^H COSY, TOCSY, NOESY, and HMBC correlations for pipecolisporin (**1**).

**Figure 3 pharmaceuticals-14-00268-f003:**
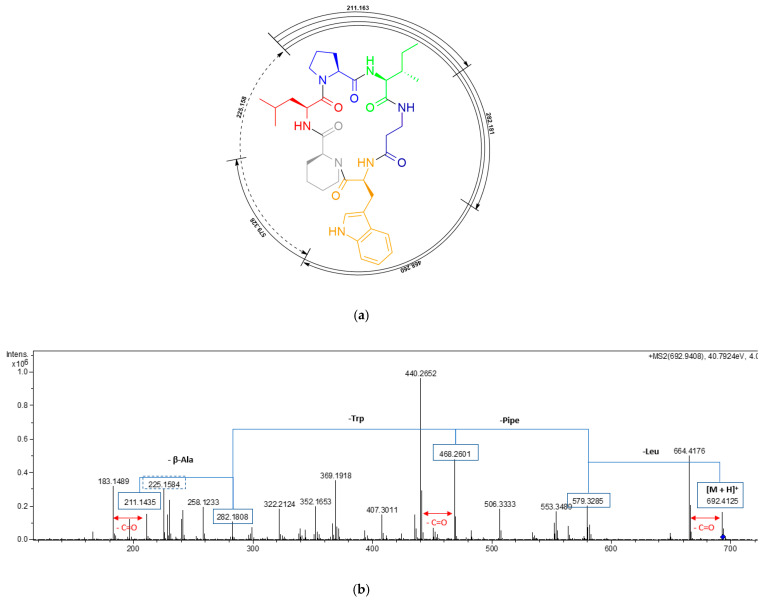
ESI-TOF MS/MS fragmentation of pipecolisporin (**1**). (**a**) Relevant fragments found. (**b**) MS/MS spectrum with the most relevant fragments highlighted.

**Table 1 pharmaceuticals-14-00268-t001:** NMR data of pipecolisporin (**1**) in DMSO-*d_6_*.

Amino Acid	Position	δH, m, J (Hz)	δC, Mult	HMBC (H to C)	NOESY
Pro	1	3.99, m	62.9, CH	CO Pro	NH Ile
	2	2.19, m, 1.77, m	29.4, CH_2_	CO Pro	
	3	1.99, m, 1.84, m	25.7, CH_2_		
	4	3.72, m 3.56, m	47.3, CH_2_		Leu 1
	CO		171.8, C		
Ile	NH amide	6.47, d, (8.0)		Pro 1	
	1	4.01, m	57.5, CH		NH β-Ala
	2	1.91, m	37.0, CH		
	3	1.30, m, 1.08, m	24.7, CH_2_		
	4	0.82, m	12.1, CH_3_	Ile 3	
	2-Me	0.81, m	16.2, CH_3_	Ile 3	
	CO		171.2, C		
β-Ala	NH amide	7.05, m		CO Ile	Ile 1
	1	2.43 m, 2.11, m	35.2 CH_2_		
	2	3.71, m, 3.12, m	35.0, CH_2_		NH Trp
	CO		171.0, C		
Trp	NH amide	8.56, d, (8.9)			
	1	4.49, m	51.8, CH	Trp 2	
	2	3.06, d, (14.8) 3.17, dd, (14.8, 4.1)	27.6, CH_2_	Trp 1, Trp 3	
	3		108.9, C		
	4	7.29, d, (2.0)	125.0, CH		
	NH	11.00, bs		Trp 5	
	5		136.7, C		
	6	7.34, d, (8.1)	118.7, CH		
	7	7.06, m	121.7, CH		
	8	6.97, m	108.9, CH		
	9	7.44, m, (8.0)	118.7, CH		
	10		127.7, C		
	CO		170.9, C		
	1	3.66, m	56.7, CH	Pipe 5	NH Leu
	2	1.45, m, −0.70 m	23.6, CH_2_		NH Leu
Pipe	3	0.99, m, 0.78, m	20.7, CH_2_		
	4	1.25 m, 0.50 m	23.9, CH_2_		
	5	4.32 m, 2.10 m	39.4, CH_2_	CO Trp	
	CO		169.6, C		
	NH amide	8.95, d, (8.5)		CO Pipe	
	1	4.53, m	50.3, CH		Pro 4
	2	1.85, m, 1.53, m	38.3, CH_2_		
Leu	3	1.62, m	25.2, CH	CO Leu	
	4	0.81, m	20.6, CH_3_		
	4’	0.90, m	23.9, CH_3_		
	CO		173.5, C		

## Data Availability

The data presented in this study are available on request from the corresponding author.

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
