# Peer review of "Pipecolisporin, a Novel Cyclic Peptide with Antimalarial and Antitrypanosome Activities from a Wheat Endophytic Nigrospora oryzae"

_pharmaceuticals, 2021, doi:10.3390/ph14030268_

Round 1
Reviewer 1 Report
Manuscript Number: pharmaceuticals-1138124
entitled: Pipecolisporin, a novel cyclic peptide with antimalarial and an-titrypanosome activities from a wheat endophytic Nigrospora oryzae
I had great pleasure reviewing this article. This is a well-conducted scientific study, done thoroughly and expressed concisely. Therefore, the manuscript is suitable for Pharmaceuticals after considering the below comments:
- Please add some general information about the term “hexapeptide” and its biological activity. Why is pipecolisporin different?
Author Response
- Please add some general information about the term “hexapeptide” and its biological activity. Why is pipecolisporin different?
We acknowledge the positive comments of this reviewer. Please note that the term hexapeptide makes reference to the type of compound isolated, a peptide with six amino acid units. There is nothing special in this kind of compounds that makes them remarkable form the biological point of view and it is only the particular structure and biological activity of individual molecules of this class what makes them interesting. In the particular case of pipecolisporin, the interest comes from its novel structure combined with the remarkable antiparasitic activity displayed.
Reviewer 2 Report
The manuscript entitled “Pipecolisporin, a novel cyclic peptide with antimalarial and antitrypanosome activities from a wheat endophytic Nigrospora oryzae” is about to evaluate antitrypanosome activity of Pipecolisporin.
Experiments were well developed and done but this manuscript need modification to improve the quality for publication.
Line 58, “.” Must be removed.
I am not sure whether “Compound is good for the Pipecolisporin, so if authors do not have specific reason to use the “compound”, please consider the compounds with cyclic peptide again. Authors already use that at the title, so I think compound is not unnecessarily confusing word for Pipecolisporin.
Line 152, It is not clear what the “Berk. & Broome” mean? so authors need clarify that.
At the 3.4. Isolation and Identification of Pipecolisporin section, Authors used several purification systems to isolate the Pipecolisporin, but it is not clear which fraction actually showed anti-malarial activity. For example, did authors follow the fraction that possessed anti-malarial activity or just have focused to find the Pipecolisporin?
Line 177, “:” should not be italic style.
At 3.5.4. Antimicrobial Aassays, Scientific name must be italic style and there are 2 at aassays.
At discussion section, authors talk about the IC50 of benznidazole. Unfortunately, I could not see any graph of that, so my question is the result was made by authors or referenced from other paper? I request that authors make sure that because if the IC50 of benznidazole is referenced from other paper, it is not comparable with IC50 of benznidazole, but if both of activity were checked by authors, the results need to be provided.
At reference, Authors need double check the style. I found “:” after first name, missing the italic style for scientific name, and sometimes capital letters for all of the word used for reference title.
Author Response
We acknowledge the positive comments of the reviewer. We have copied his/her concerns and answers to them are below each paragraph:
Line 58, “.” Must be removed.
Please note that sp. is the abbreviation for species and the dot cannot be removed in this case
I am not sure whether “Compound is good for the Pipecolisporin, so if authors do not have specific reason to use the “compound”, please consider the compounds with cyclic peptide again. Authors already use that at the title, so I think compound is not unnecessarily confusing word for Pipecolisporin.
Compound is a word commonly used in articles describing the structure of new molecules. It is alternated with others such as molecule, pipecolidepsin, etc. in the article to not use always the same term to refer to the new molecule isolated
Line 152, It is not clear what the “Berk. & Broome” mean? so authors need clarify that.
Please note that Berk. & Broome is the taxonomic authority and is part of the taxonomy of the fungal strain. It is the name of the scientist of scientists who first published the name of the species.
At the 3.4. Isolation and Identification of Pipecolisporin section, Authors used several purification systems to isolate the Pipecolisporin, but it is not clear which fraction actually showed anti-malarial activity. For example, did authors follow the fraction that possessed anti-malarial activity or just have focused to find the Pipecolisporin?
As stated in the article, the purification of the compound was guided by its chemical novelty and once purified, it was tested against a panel of microorganisms, cancer cell lines and parasites, displaying good inhibitory properties against P. falciparum and T. cruzi.
Line 177, “:” should not be italic style.
This has been corrected
At 3.5.4. Antimicrobial Aassays, Scientific name must be italic style and there are 2 at aassays.
This has been corrected
At discussion section, authors talk about the IC50 of benznidazole. Unfortunately, I could not see any graph of that, so my question is the result was made by authors or referenced from other paper? I request that authors make sure that because if the IC50 of benznidazole is referenced from other paper, it is not comparable with IC50 of benznidazole, but if both of activity were checked by authors, the results need to be provided.
We acknowledge this comment from the reviewer. Benznidazole and chloroquine were included as controls in the T. cruzi and P. falciparum assays, respectively. We have included in the supporting information the curves obtained for both compounds in our assays.
At reference, Authors need double check the style. I found “:” after first name, missing the italic style for scientific name, and sometimes capital letters for all of the word used for reference title.
All references have been checked. Please note that we include capital letters in all the words of the title in those cases where the title of the article is written in that way in the original article. We have modified this point in some references to be compliant with the original publication in all cases. All the modifications in the style of the references have been highlighted with the Track Changes option of Word.